

# Dissolved organic carbon mobilized from organic horizons of mature and harvested black spruce plots in a mesic boreal region

Keri Bowering[1], Kate A. Edwards[2], Xinbiao Zhu[2], Susan E. Ziegler[1]

[1]Earth Sciences, Memorial University of Newfoundland, St. John's, A1C 5S7, Canada

[2]Canadian Forest Service, Natural Resources Canada, Ottawa, K1A 0E4, Canada

*Correspondence to*: Keri Bowering (klbowering@mun.ca)

**Abstract.** Boreal forests are subject to a wide range of temporally and spatially variable environmental conditions driven by seasonal and regional climate variations, in addition to disturbances such as forest harvesting and climate change. Among the various ecological mechanisms affected by disturbance, is the transport rate of dissolved organic carbon (DOC) from surface soil organic (O) horizons to deeper mineral SOC pools and the adjacent aquatic systems. Here, we examine the transport of DOC from surface O horizons across a boreal forest landscape using passive pan lysimeters in order to identify controls and hot moments of DOC mobilization from this key C source. To do so, we specifically addressed (1) how DOC fluxes from O horizons vary on a weekly to seasonal basis in both forest and harvested plots, and (2) how soil temperature, soil moisture and water inputs relate to DOC fluxes in these plots over time. The total annual DOC flux from O horizons was greater in the warmer harvested plots than in the forest plots (54 g C m$^{-2}$ vs 38 g C m$^{-2}$ respectively; p=0.008), despite smaller aboveground C inputs and smaller SOC stocks in the harvested plots. Water input, measured as rain, throughfall and/or snowmelt depending on season, was positively correlated to temporal variations in soil water and DOC fluxes. Soil temperature was positively correlated to temporal variations of DOC concentration ([DOC]) of soil water and negatively correlated with water fluxes, but no relationship existed between soil temperature and DOC fluxes. Soil moisture was negatively correlated to temporal variations in [DOC] in the harvested plots only.

The relationship between water input to soil and DOC fluxes was seasonally dependent in both plot types. In summer, a water limitation on DOC flux existed where weekly periods of no flux alternated with periods of large fluxes, suggesting that increased water fluxes over this period would result in proportional increases in DOC fluxes. In contrast, a flushing of O horizons occurred during increasing water inputs and decreasing soil temperatures in autumn, prior to snowpack development. Soils of both plot types remained snow-covered all winter, which protected soils from frost and limited winter soil water fluxes. The largest water input and soil water fluxes occurred during spring snowmelt, but did not result in the largest fluxes of DOC, suggesting a production limitation on DOC fluxes over both the wet autumn and snowmelt periods.

While future increases in annual precipitation could lead to increased DOC fluxes, the response may be dependent on the intra-annual distribution of this increase. Increased water input during the already wet autumn, for instance, may not lead to increased fluxes if the DOC pool is not replenished. Potential reductions in snow cover, however, leading to a reduction in soil insulation



and increased occurrence of soil frost in addition to increases in winter-time water fluxes, could be an important mechanism of increased DOC production and fluxes from O horizons in winter.

## 1 Introduction

Boreal forests occupy 11 % of the total land surface thus spanning a variety of topographies and climate zones (Bonan and Shugart, 1989), and contain organic matter rich soils that store approximately 19% of the global soil organic carbon (SOC) pool (Pan et al., 2011). Through various biological, chemical, and physical processes, SOC can be mobilized as $CO_2$ to the atmosphere, as dissolved organic carbon (DOC) to deeper SOC pools, or as DOC to groundwater and surface waters. Boreal forest heterotrophic soil respiration (R), representing approximately 40% of boreal forest gross primary production (Luyssaert

et al., 2007), is a significant fate of boreal forest SOC. Eddy covariance measurements of $CO_2$ exchange, used to estimate C balance or net ecosystem production (NEP), capture soil R. In contrast, boreal forest SOC mobilized as DOC and transported outside the reach of the eddy covariance footprint is not accounted for in NEP estimates of boreal forests, despite the potential for constituting a significant portion of NEP (Gielen et al., 2011). The exclusion of DOC fluxes in measurements of NEP therefore potentially leads to significant underestimates of the boreal forest C balance.

The importance of upland forest (ie. non-wetland) SOC as a source of DOC to boreal forest surface waters is likely quite variable among boreal regions, due to large variations in topography and precipitation (McGlynn and McDonnell, 2003). In lower relief catchments, for example, SOC of the riparian zone represents an important DOC source to streams (Ledesma et al., 2017), whereas the SOC mobilized from upland forest soil in these areas may be lost via respiration and/or sequestered within deeper mineral soil pools before reaching the inland water network. Higher relief catchments are examples where upland

forest soils are likely much more connected to the aquatic zone, especially during large precipitation events (Raymond and Saiers, 2010), and/or periods of the year when the water table is high (Laudon et al., 2011; Schelker et al., 2013). Therefore, the importance of the upland forest SOC contribution to the aquatic zone is likely not generalizable across boreal forest ecosystems, constituting examination within specific regions and under varying environmental conditions (Marín-Spiotta et al., 2014).

The large range in values of organic (O) horizon DOC fluxes reported from field studies in temperate and boreal forest systems (3–122 g C m$^{-2}$ at 5 cm depth, Neff & Asner, 2001; 10–40 g m$^{-2}$ y$^{-1}$, Michalzik et al., 2001), are presumably due to both real variability, and variability associated with different methodologies used. Real-world variability is expected given the known spatial heterogeneity of soil and hydrological aspects of forests (Creed et al., 2002), and could also be attributed to seasonal and inter-annual variability of temperature and precipitation in forests spanning the boreal biome (Bonan and Shugart,

1989). The spatial and temporal variability of these factors both within boreal forests and among them, and the laborious and complicated nature of measuring DOC fluxes *in situ*, points to a need for additional information enabling a process-based understanding of these fluxes that would facilitate scaling up to multi-year regional, catchment and biome-scale estimates. The



current lack of empirical information about belowground C processes, such as the rate of DOC transfer and how it is regulated, continues to prevent accurate representation in SOC models (Stockmann et al., 2013) and forest C balance models (Kurz et al., 2009; Smyth et al., 2013).

Soil DOC is often extracted either in the laboratory or in the field using tension lysimeters and the variations in extractable DOC concentration in these studies are used to understand DOC dynamics and environmental controls on DOC production. The controls on extractable soil DOC include soil temperature and moisture (Christ and David, 1996; Moore 2008), pH (Andersson et al., 2000), soil C:N (Godde et al., 1996) and DOC bioavailability (Cleveland et al., 2004), but many of these are not confirmed in field settings.  In terms of DOC fluxes *in situ*, the physical effects of hydrology are thought to be more important than biological controls, although clarification of the water flux-DOC flux relationship was suggested as an area of further research (Karsten Kalbitz et al., 2000; Neff and Asner, 2001). More recent field studies have focused on specific hydrological controls, such as annual throughfall inputs (Klotzbücher et al., 2014), soil drying followed by rewetting (De Troyer et al., 2014), soil frost (Haei et al., 2010), and snowmelt (Finlay et al., 2006). It is likely that a combination of these physicochemical and environmental factors regulate DOC production and mobilization through soil, but the relative importance of each of these factors is, in part, dependent on the scale of investigation, both spatially and temporally (Michalzik et al., 2001). On larger spatial scales, climate transect studies within the boreal forest zone have revealed greater DOC fluxes at warmer (low-latitude) relative to cooler (high-latitude) sites, suggesting that this difference is explained by higher N deposition (Kleja et al., 2008) and/or higher net primary productivity (Fröberg et al., 2006; Ziegler et al., 2017) in the lower latitude sites. Furthermore, DOC fluxes from O to mineral horizons in white pine stands was observed to be negatively correlated with stand age, (Peichl et al., 2007), and a stand species comparison study demonstrated larger DOC fluxes from the thicker O horizons of Norway spruce stands relative to silver birch stands (Fröberg et al., 2011).

Black spruce trees are one of the dominant species in North American boreal forests (van Cleve et al., 1983) and these forests span a wide range of environmental conditions that drive large variations in SOC decomposition (Wickland et al., 2007) and SOC persistence across sites (Schmidt et al., 2011). Here we exploit both spatial variations (plot type) and temporal variation (weekly to seasonal) in environmental conditions to investigate the drivers of DOC fluxes from O horizons of a maritime boreal black spruce hillslope site that receives moderately high annual precipitation ($\sim$ 1000mm yr$^{-1}$), approximately half as snow. The objectives of this study were: 1) to measure within year variability of DOC fluxes over one year from O horizons of podzols in two contrasting boreal plots that are typical of the managed boreal forest landscape and 2) to utilize the different environmental conditions observed in the two plot types and across seasons in order to understand the role of varying environmental conditions in regulating DOC fluxes from O horizons.

## 2 Materials and Methods

### 2.1 Site Description



This study was conducted in an experimental forest harvest site within a mature black spruce forest at the Pynn's Brook Experimental Watershed Area (PBEWA) located near Deer Lake, western Newfoundland and Labrador, Canada. (lat. 48° 53'14", long. 63° 24' 8"). The region receives approximately 1095 mm of precipitation annually with a mean annual temperature of 3.6 °C (Environment Canada Climate Normals, Deer Lake Airport 1981-2010). The site consists of 2 hectares

divided into eight 50 x 50 m plots. Four of the plots were left un-harvested and four were randomly selected for clear-cutting. The four clear-cut plots were harvested on July 07-10, 2003 using a short-wood mechanical harvester, with minimal disturbance to the underlying soil and with any deciduous trees left standing. Further information on site preparation and conditions can be found in Moroni et al., 2009. The harvested plots were not replanted following clear-cutting and had naturally recovered moss, herb and shrubbery by the time of sampling for this study, but the regeneration of conifers remains scarce.

The 10-year post harvest plots will be referred to as *harvested plots* and the mature 80-year-old black spruce plots will be referred to as *forest plots* throughout.

Soils were classified as humo-ferric podzols with morainal parent material by Moroni et al., 2009. Furthermore, O horizon soil was sampled specifically for this study by taking three 20 x 20 cm samples from three forest plots and three 20 x 20 cm

samples from three harvested plots (n = 9 for each type). Living vegetation was removed, the thickness of each sample was measured, and the soil was sieved using a 6 mm sieve and dried at 50 °C for 48 hours. Samples were ground using a Wiley mill and subsampled for elemental analysis on a Carlo Erba NA1500 Series II Elemental Analyser (Milan, Italy) at Memorial University of Newfoundland. These samples were used to determine soil % C, soil C stock (kg C m$^{-2}$). Mineral soil was sampled below each O horizon sample with a soil corer (length: 15 cm, diameter: 5.5 cm). Each mineral soil sample was sieved

using a 2 mm sieve and dried at 50 °C for 48 hours and weighed. Once dried and weighed, samples were ground using a ball mill and subsampled for elemental analysis as above for O horizon samples. The % rock fragments (>2mm) by volume was estimated using Eq. (1):

Z2 = Z1 (2-Z1)  (Brakensiek and Rawls, 1994)


where Z2 = % rock by volume

Z1 = % rock by weight

Bulk density of O and mineral soils was calculated using the volume and dried mass of the soil sample. Litterfall was collected

using four 0.34 m$^2$ litter traps placed on the forest floor in four plots per plot type from August 2012 to August 2013. Litter was collected in early spring and late fall, sorted into needles, bark, cones, lichen and deciduous leaves, dried at 60 °C over 48 hours, and weighed. A litterfall C input was estimated (Table 1) by applying concentrations of 542 mg C g$^{-1}$ for both twigs and needles and 552 mg C g$^{-1}$ for cones of black spruce litter fall (Preston et al., 2006).



## 2.2 Lysimeter Installation and Sample Collection

Passive pan lysimeters were installed at the interface between the O and mineral horizon. Each lysimeter has a 0.12 m² footprint and collects water percolating through the O horizon with a maximum solution collection capacity of 25 L. The lysimeters were designed using reported recommendations for achieving accurate volumetric measurements of soil leachate (Radulovich and Sollins, 1987; Titus et al., 1999). It was desirable for this study that: 1) the collection pan was filled with glass beads to mimic the mineral horizon and reduce preferential flow into the lysimeter, 2) the collection pan directs leachate immediately into a deeper storage container, avoiding potential issues of sample evaporation from the collection pan, and 3) the buried storage reservoir was placed away from the collection pan so that soil and snowpack directly above and upslope from collection area was not disturbed during sample collection.

Installation of lysimeters began in July 2012 and was completed the following spring in May 2013.  Four lysimeters were installed in three plots of each plot type for a total of 12 forest lysimeters and 12 harvested lysimeters. Collection began on July 12, 2013 from forest and harvested lysimeters. Synchronized sampling from lysimeters of both plot types was carried out every 7 to 15 days from July 5th, 2013 to December 31st, 2013, once between January 1st and April 1st 2014, and every 7 to 15 days from April 01 2014 to July 23, 2014. Lysimeter samples were stored at 4°C immediately following collection, were filtered using pre-combusted GF/F-size Whatman filter paper, preserved with mercuric chloride within 24 hours of collection, and stored at 4°C in the dark until analysis.

Lysimeter collection efficiency testing was completed on 3 forest lysimeters and 3 harvested lysimeters following the study period. The soil on top of and around the lysimeter catchment area was first saturated, and then the area directly above each lysimeter was watered uniformly with 10 L of water and the volume of solution collected by the lysimeters was retrieved. This was repeated 3 times on each of the lysimeters to determine the efficiency of the lysimeter system in collecting the leachate from the footprint of organic soil directly above the installed pan.  Lysimeter efficiency was found to be 92.3 ± 21 % and 88.6 ± 18 % in the forest and harvested plots, respectively. No statistically significant difference between the collection behavior of the forest and harvested forest plot lysimeters was detected (t-test; p=0.8248).

## 2.3 Environmental Monitoring

Three soil temperature and moisture probes per plot type (Decagon ECH₂O -TM) were installed mid- organic horizon at approximately 5-cm depth, and two were installed in the mineral layer at approximately 15-cm depth. These probes measure the dielectric constant of the soil using capacitance/frequency domain technology, providing % volumetric water content (VWC), however these probes were not specifically calibrated to these soils so values are best interpreted as relative rather than absolute measures of VWC, especially for the O horizons. Handheld spot measurements using a HydroSense II Soil Water Content Reflectometer on selected days (data not shown), confirmed the consistently wetter O horizons in the harvested plots



as measured by field probe measurements (Fig. 1; Table 1). The hydraulic properties of the O horizon soil was determined for 4 forest and 4 harvested plot samples (Supplementary Table 1) using an automated HYPROP system (Decagon/UMS), which measures water potential to generate moisture release curves.  The field capacity ranges of 14.1 – 21% VWC in the harvested plots and 12.6 – 19.3% in the forest plots determined by HYPROP analysis are in upper range of values measured with the

Decagon field probes (Fig. 1) suggesting that the field probes are measuring within the range of field capacity values.

One tipping bucket rain gauge (RST Instruments Model TR-525) was installed in an open area on site to monitor local rain and air temperature. Data from this tipping bucket were compared with regional rainfall and air temperature reported by Environment Canada at the Deer Lake Airport (lat. 49°13'00" N, long. 57°24'00" W) approximately 50 km away, and were

found to be well correlated ($R^2$= 0.882, p<0.0001).  Regional data from the Deer Lake Airport was used to fill a gap in our onsite daily rainfall and mean daily air temperature data between July 7th and 24th, 2013. Snowmelt water input was estimated using changes in snow depth between each lysimeter collection day measured near each lysimeter in both the forest and harvested plots. The average snow depth change by plot type was multiplied by an estimated maritime snow density of 0.343 g cm$^{-3}$ (Sturm et al. 2010) to provide an estimated snowmelt water input value. We acknowledge that snow density is variable

both within the snow profile and over the course of snowmelt and that this calculation only provides a rough estimate of the water input to the soil from snowmelt. These estimates were combined with rainfall where applicable to give a total *water input* to the forest floor over each collection period for comparison with the water fluxes independently measured across the O to mineral horizon interface by the lysimeters.

A snow pit was analyazed for each plot type in April 2014 just prior to the onset of snowmelt. A series of 15 cm long snow cores were collected beginning from the top of the snowpack down to the forest floor to obtain a sample of the entire snowpack per plot type. All snow cores per plot type were combined and measured to provide a mean DOC concentration in the snow of forest and harvested plots. The snow depth of each plot, combined with the estimated snow density (0.343 g cm$^{-3}$) and DOC concentration was used to determine a snow DOC input to the forest floor (Table 1). Throughfall was collected on an event

basis using 10 buckets (0.36 m$^{-2}$ collection area) distributed within a 50 x 50 m forest plot in May, August and October 2015. Synchronized collection of open rainfall using 5 buckets was completed in an adjacent harvested plot. Prior to the first sampling date a preliminary variability experiment was conducted in October 2015 onsite to determine the most practical number of buckets required to capture the variability within forest and harvested plots. Forty buckets were installed in a forest plot and ten in a harvested plot and left out for one rain fall event. The contents of each bucket was sampled, filtered and analyzed for

DOC concentration. From these data a Monte Carlo simulation was used to predict the relationship between number of buckets deployed and the variability of DOC concentration captured. It was found that installing ten buckets in the forest plots, and five in the harvested plots captured a similar amount of variation in water volume and DOC concentration as deploying forty gauges in the forest plot and ten in the harvested. Mean DOC concentrations of each collection was determined for each





collection period and used as a seasonal representation of throughfall and open fall DOC concentrations. Seasonal DOC was then scaled up to an annual DOC input estimate (Table 1) using measured annual open rainfall and throughfall water inputs.

## 2.4 Soil Respiration

Measurements of soil respiration were made at biweekly intervals for the snow-free growing seasons (May–November) in 2013-2015. Four collars consisting of a 7-cm section of 10-cm inside diameter PVC pipe were inserted into the ground 8 months prior to the start of measurement in four forest plots and four harvested plots. Soil respiration rate and soil temperature were measured using a LI-6400-09 soil chamber and a penetration soil temperature probe, both attached to LI-6400 portable $CO_2$ infrared gas analyzer (IRGA). Volumetric soil water content was measured with a Campbell Hydro-Sense penetration probe inserted in the soil to the depth of 10 cm in the vicinity of the PVC collars.

## 2.4 Chemical Analysis and Flux Calculation

The DOC concentration of each lysimeter sample, as well as throughfall, rain and snow samples, was measured using a high-temperature combustion analyzer (Shimadzu TOC-V). The measured DOC concentration, the total volume collected by lysimeters, the number of collection days, and the lysimeter collection area were used to calculate a DOC flux (g C m$^{-2}$ d$^{-1}$). Water flux was calculated using the measured lysimeter volume on each collection day and the lysimeter collection area (L m$^{-2}$ d$^{-1}$).

## 2.4 Statistical Analyses

All statistical analyses were performed using RStudio version 1.0.136. T-tests were used to determine plot type differences in mean annual soil moisture and soil temperature. ANOVAs were used to determine plot treatment differences in total annual DOC flux, water flux and DOC concentration, mean organic horizon thickness, mean organic and mineral soil % C, mean organic and mineral soil C stocks, and mean annual litterfall between forest plots and harvested plots over the entire study period (Table 1, Supplementary Table 1, Fig. 2; *asterisks*). A repeated measures linear mixed effects (RM-LME) model was used to assess the effects of time, and the interaction between time (collection day) and plot type on the intra-annual variation of DOC fluxes, water fluxes, and DOC concentration (Supplementary Table 2). Post-hoc Tukey tests were used to determine significant differences in DOC flux, water flux and DOC concentration between forest and harvested forest plots on individual collection days (Fig. 1d-e; *asterisks*). The data were grouped into three seasons: summer, autumn and spring snowmelt, and a two way ANOVA was used to assess the effects of water input, season and their interaction on DOC fluxes (Table 3).

Correlation testing was used to assess the relationships among data from lysimeter collections (DOC flux, water flux and DOC concentration) and mean soil temperature, mean soil moisture and daily water input (Table 2) across 30 collection days. Multiple regressions were not used due to the multi-collinearity of many of the predictor variables, which affected the estimated



regression parameters. Individual correlations, however, were assessed to evaluate the strength of relationships among variables within the dataset.

A linear mixed effects model was used to examine the effects of plot type, sample year (2013–2015), and their interactions on soil respiration. The interaction term was further analysed with a post-hoc least square means test. Linear interpolation was used to calculate cumulative soil respiration for the snow-free growing season during the period of 2013–2015. A multiple linear regression was used to explain the dependence of soil respiration on soil temperature, moisture and the soil temperature by soil moisture interaction.

## 3 Results

### 3.1 Environmental Conditions

The regional mean annual air temperature over the July 2013 to July 2014 study period was + 3.4 °C (range: - 21.0 °C to + 22.7°C), and 1305 mm of total precipitation fell, including 483 cm of snowfall. Comparison to climate normal estimates derived from 1981-2010 indicate the region received 210 mm more total precipitation and 50 cm more snowfall over the study period than average, and that the mean air temperature was 0.2 °C cooler than average. The greatest total rainfall occurred over the autumn period (376.2 mm), followed by the summer (291.7 mm), snowmelt/spring (121.9 mm) and then winter (101.6 mm). Summer was the only period to experience 10 consecutive days of <10mm/day of rainfall (Fig. 1; *shaded areas*). Throughfall in the forest plots was approximately 60% of the incoming bulk rainfall measured in the open harvested plots. The greatest total snow fall occurred during the winter period (440.6 cm). Total autumn snowfall was 36.7 cm, and snowmelt/spring snowfall was 8.4 cm, with no snow falling in the summer. The snowpack depth just prior to snowmelt was 83 cm in the forest plots and 110 cm in the harvested plots.

The O horizons in the harvested plots were generally warmer than those in the forest plots (Table 1, Fig. 1B; forest plot range: +1.1°C to +16°C; harvested plot range: +1.4°C to +19.7°C). In summer, soil temperatures maintained an approximate 2°C difference. Decreasing air temperature in the autumn was associated with a convergence of soil temperature such that winter soil temperatures in the two different plot types were similar. Increasing air temperatures in the spring were again accompanied by a divergence of soil temperatures between the two plot types (Fig. 1B). The snowpack persisted throughout winter such that the soils in both plot types were insulated against freezing, with a partial melt occurring from January 14 to January 20, 2014. Soil temperatures began increasing in the spring about two weeks earlier in the harvested plots than in the forest plots,



indicating an approximate two- week lag in the snow free period in the forest plots compared to the harvested plots (Fig. 1B; *snowpack*).

The O and mineral horizons were consistently wetter in harvested plots relative to the forest plots over the duration of the study (VWC annual means of 20.2 and 13.2; Fig 1c), but given the high variability and few measurement replicates (n=3 O horizon, n=2 mineral horizon) this pattern was not statistically confirmed (Table 1). The O horizons experienced long periods of drying in the summer, especially in July 2013 (Fig. 1B; *shaded areas*) but there was little change in soil moisture over the winter other than during a short episode of warming and snowmelt in January 2014, which raised VWC of O horizons in both plots types for approximately 2 weeks.

**3.2 Soil Respiration**

The temporal range in instantaneous $CO_2$ efflux rates during the lysimeter measurement period (July 2013 – July 2014; Fig. 1A) was approximately 2 – 4.8 g C m$^{-2}$ d$^{-1}$ in the forest and harvested plots, amounting to 672.2 and 711.9 g C m$^{-2}$ y$^{-1}$ in the forest and harvested plots, respectively. Highest efflux rates occurred in the summer and decreased to lowest values in autumn in both plot types. Lowest rates occurred following snowmelt and increased in both plot types as soils warmed.

There was no overall significant difference in soil respiration between plot types for the 2013- 2015 growing season estimates however, there was a significant plot type by sample year interaction effect on soil respiration (Supplementary Table 3). The multiple comparisons found that soil respiration in the harvested plot was lower relative to that in the forest plot for 2014 and 2015 growing seasons, but not 2013 (Supplementary Table 4 and 5). Soil respiration was positively related with soil temperature but negatively related with soil moisture content, and the presence of a soil temperature by soil moisture interaction on soil respiration in the regression analysis indicated the effects of soil temperature on soil respiration had been modified by soil moisture (Supplementary Table 6).

**3.3 Soil Properties and Aboveground Litterfall**

Soil bulk density was not different between the forest and harvested plots for either O or mineral soil horizons (Table 1). However, O horizon depth was almost twice as great in the forest plots compared with the harvested plots (means of 8.17 cm and 4.26 cm respectively; Table 1), corresponding to an estimated 78% greater SOC stock in forest plots relative to harvested plots (2.39 kg C m$^{-2}$ and 1.34 kg C m$^{-2}$; Table 1). Annual litterfall inputs to the soil surface were greater in the forest plots (256.5 g m$^{-2}$ y$^{-1}$ and. 11.8 g m$^{-2}$ y$^{-1}$), amounting to an estimated 212.4 g C y$^{-1}$ and 12.4 g C y$^{-1}$ reaching the forest floor as litterfall in the forest and harvested plots respectively (Table 1).



### 3.4 DOC Concentration

The mean annual volume weighted DOC concentration collected by lysimeters was 29.4 and 26.1 mg C L$^{-1}$ in the forest and harvested plots (Fig. 2a) were not found to be statistically different (p=0.09). The mean annual DOC concentration was volume weighted because lysimeter collections were not made at even time intervals throughout the year. Seasonal ranges of absolute

concentrations include summer mean concentrations of 55 and 45 mg C L$^{-1}$, autumn means of 42 and 38 mg C L$^{-1}$, winter means of 18 and 13 mg C L$^{-1}$ and spring snowmelt means of 25 and 20 mg C L$^{-1}$ in the forest and harvested plots, respectively. The DOC concentration also varied with collection day and exhibited an interaction of collection day by plot type; higher DOC concentrations were measured in forest plots relative to the harvested plots in 9 of 25 sampling times most commonly observed during summer and early autumn.  No differences in DOC concentration were detected between plot types during the late

autumn and winter (October to April; Fig 1d). One difference was detected in early spring (April 15$^{th}$) and differences were observed into the following summer.  Much of the intra-annual variation in DOC concentration was explained by soil temperature (positive correlation; Table 3) and water flux variation (negative correlation; Table 3) in both plot types. However, in the harvested plot only, DOC concentration was negatively correlated with soil moisture.

The mean DOC concentration within the snow profile immediately prior to snowmelt was 7.5 mg C L$^{-1}$ and 3.3 mg C L$^{-1}$ in the forest and harvested plots, respectively, corresponding to a total depth of 84 cm and 110 cm, which amounted to a potential DOC input to the soil of 2.1 g C m$^{-2}$ and 1.2 g C m$^{-2}$ over the course of snowmelt in the forest and harvested plots, respectively (Table 1).  The mean DOC concentration in throughfall measured in one forest plot was approximately 7 mg DOC L$^{-1}$ and open rainfall measured in one adjacent harvested plots was approximately 3 mg DOC L$^{-1}$, consistent across May, June and

October samples. Scaling up to an annual DOC input estimate to soil using annual rainfall amounted to 5.5 g m$^{-2}$ and 3.9 g m$^{-2}$ in the forest and harvested plots, respectively (Table 1).

### 3.5 Water and DOC Fluxes

The mean annual O horizon water flux was 2040 L m$^{-2}$ (+/- 129) in the harvested plots and 1366 L m$^{-2}$ (+/- 344) in forest plots, revealing a 49% greater flux of water through the O horizons in the harvested plots relative to the forest plots (Fig. 2b; p=

0.0357). This corresponded to DOC fluxes of 54 g C m$^{-2}$ (+/- 3) and 38 g C m$^{-2}$ (+/- 5) in the harvested and forest plots, respectively, representing a 30% greater annual loss of DOC from the O horizon of harvested plots (Fig. 2c, p=0.00836). The intra-annual DOC and water fluxes varied with collection day, with an interactive effect of plot type and collection day on both fluxes (Table 2a,b). Water fluxes were generally greater in harvested plots than forest plots on a given collection day, often corresponding to greater DOC fluxes in harvested plots (Fig. 1d,e; asterisks). The difference in water flux between plot

types was significant on 8 of 30 collection days while the difference in DOC flux between plot types was significant less often (6 of 30).





Longer periods of soil drying and low rainfall, occurring predominately during summer, corresponded with periods of little to no water flux and, consequently, little to no DOC flux in both harvested and forest plots (Fig. 1b,d,e; *shaded areas*). In contrast, periods of relatively high moisture and consistent rainfall, occurring predominately in autumn, corresponded with high and consistent water and DOC fluxes. During spring snowmelt, however, when the DOC concentration was relatively low, large

water fluxes did not result in the largest fluxes of DOC (Fig. 1; April 08 2014 to May 01 2014). The highest DOC flux over the study period was observed in early autumn when a large rain event followed a warm period of soil drying. Soil water fluxes were negatively correlated with soil temperature (Table 2a) and there was a strong positive correlation between water input and both soil water and DOC fluxes measured in both plot types (Table 2c).There was an interaction between season and water input on DOC fluxes (Table 3), where a more direct relationship between water input and DOC fluxes observed in the summer

(Fig. 3a), but DOC fluxes exhibited a tapering off in autumn and snowmelt when water input was high (Fig. 3b,c).

## 4 Discussion

### 4.1 DOC flux is an important component of the annual boreal forest C balance

The annual DOC flux from O horizons across a black spruce dominated site was 38 g C m$^{-2}$, approximately 5% of the $CO_2$ efflux from the surface of the same soils measured during the snow-free season (712 g C m$^{-2}$). This is a larger flux of DOC

than that reported for balsam fir dominated forests, where the annual flux was approximately 10 - 30 g C m$^{-2}$ using similar lysimeters in the same region (western Newfoundland and southern Labrador, Canada), which corresponded to less than 5% of measured soil $CO_2$ efflux (Ziegler et al., 2017). Although small relative to the total soil $CO_2$ efflux, the DOC flux values measured here are comparable to the range of annual net ecosystem productivity (NEP) from across the managed Canadian boreal forest (1990 to 2008) estimated to be between -16.2 to 55.7 g C m$^{-2}$ year$^{-1}$ (Kurz et al., 2013), supporting the notion that

mobilization of O horizon DOC can affect the C balance of a boreal forest system (Gielen et al., 2011).

Harvesting leads to sites becoming C sources to the atmosphere for several years (negative NEP), but harvested forests are expected to follow an increasing NEP trajectory during regeneration. As tree growth rates increase, forests eventually reach a compensation point where they are neither sources nor sinks of C, typically within 10-20 years following boreal forest

harvesting (Kurz et al., 2013). These estimates are based primarily on $CO_2$ efflux and biomass C sequestration with growth, but our data suggest that in the system studied here, DOC flux could also be an important component when determining this compensation point, where significant differences in DOC fluxes between forest and harvested plots are still evident 10 years after harvesting. Without knowing what proportion of this leached C is remaining in the system in deeper mineral soil horizons, we don't know how this flux is contributing to a source or sink scenario, highlighting an important area for further inquiry.

Nevertheless, the slow rate of natural forest recovery continues to affect hydrological mobilization of SOC in this black spruce site, where harvested plot values remained elevated (54 g DOC m$^{-2}$), resulting in a 30% greater mean annual flux compared to adjacent forest plots (38 g DOC m$^{-2}$). The larger DOC fluxes from the thinner soils of the harvested plots translated to an



annual loss of approximately 6% of the total SOC stock as DOC in the harvested plots, whereas 3% of the SOC stock was lost as DOC in the forest plots. This reveals a greater propensity for SOC to be mobilized as DOC in the harvested plots compared with forest plots, despite lower SOC stocks (1340 vs. 2390 g C m$^{-2}$) and inputs from aboveground litter (12.4 vs. 212.4 g C m$^{-2}$ year$^{-1}$). Increases in DOC fluxes from O horizons immediately following and up to 5 years after boreal forest harvesting have

been previously documented (Pirainen et al., 2002; Piirainen et al., 2007; Kalbitz et al., 2004), but to our knowledge this is the first study to demonstrate a longer lasting (10-year) effect in post-harvest plots.

## 4.2 O horizon DOC flux dynamics are temporally and spatially dominated by water input patterns

In both forest and harvested plots, O horizon DOC flux patterns mirrored those of water flux on a weekly to annual basis, while the contribution of DOC concentration variation to observed temporal differences is less evident in DOC flux patterns

(Fig. 1d,e,f). This is additionally described in both plot types by a strong positive relationship between water input to the forest floor (as rainfall, throughfall and/or snowmelt) and DOC flux (Table 3) with no relationship between DOC flux and soil temperature. Therefore, even across a landscape with both forest and harvested areas, characterized by different surface soil and ecosystem properties (Table 1), water input to soil is a dominant control over O to mineral horizon DOC flux dynamics on weekly to seasonal to annual time scales. Although the temporally dynamic nature of the O horizon DOC flux and the

relationship with water flux has been previously described by many lysimeter studies (reviewed by Kalbitz, 2000; Neff and Asner; 2001), terrestrial C models require continued validation of belowground C processes from field measurements to refine parameterization efforts (Smyth et al. 2013). For instance, the O to mineral horizon DOC flux is included in the Carbon Budget Model of the Canadian Forest Sector (CBM-CFS3, Kurz et al., 2008), but is parameterized using a single mean annual value of DOC flux and mean O horizon SOC stock. The close relationship between water and DOC flux shown in this study, together

with the large range in mean annual precipitation (MAP) that exists across Canada's boreal Ecoregions (173 – 1492 mm; A National Ecological Framework for Canada, 1999) and prior correlations shown between MAP and annual DOC flux across ecosystems (Michalzik et al., 2001), all suggest that DOC dynamics in models such as the CBM-CFS3 could be updated to reflect the spatial heterogeneity in MAP that exists across the boreal forest. Additionally, the increased water availability driven by tree harvesting resulted in larger DOC fluxes despite reductions in SOC stock and C inputs in this study, suggesting that O

horizon DOC fluxes could additionally be parameterized to disturbance history.

Modelling of DOC delivery to mineral soil at the landscape scale is further complicated by the fact that mobilization is not limited to vertical flow but also includes topographically and seasonally dependent horizontal or lateral flow (McGlynn and McDonnell, 2003; Tipping et al., 1997). Studies examining controls on DOC content of soils at depth focus on mineral-OM interactions and thereby employ conceptual models that are vertically dominated (e.g. Kaiser & Kalbitz, 2012). These may be

relevant in low relief landscapes or where soil hydraulic conductivity is uniformly high throughout the soil profile. However, in these mesic boreal forest podzols we observed evidence for significant horizontal flow. Passive lysimeter methods can be hydrologically disruptive and capture both vertical and horizontal soil water transport causing many lysimeter studies



concerned with vertical transport to mineral soil to model or estimate soil water fluxes instead of measuring them (Clarke et al., 2007; Fröberg et al., 2011; K. Kalbitz et al., 2004; Rosenqvist et al., 2010). In our study, on both weekly to annual timescales, more water was collected by lysimeters than the water input to soil over the lysimeter collection footprint in both plot types (56% -74% greater annual soil water flux than annual regional precipitation). Using the measured annual DOC

concentrations (Fig. 1A), this corresponds to a horizontal flow contribution of 15 g C m$^{-2}$ in the forest plots and 20 g C m$^{-2}$ in the harvested plots, potentially reducing the vertical flux to 23 and 34 g C m$^{-2}$ in the forest and harvested plots, respectively. These values are more in line with average values for boreal and temperate forests where methodology was often focused on vertical fluxes (22 g C m$^{-2}$; Michalzik et al., 2001). Horizonal water flow at the O to mineral horizon interface, created by landscape slope (Creed et al., 2013) and differences in hydraulic conductivity (Haynes and Naidu, 1998; Koch et al., 2016),

can limit the vertical entry of DOC from surface soil to mineral horizons in upper parts of the landscape, while displacing DOC to downslope terrestrial areas or directly to the aquatic zone. Therefore, horizontal flow is a potential control on the delivery of DOC to mineral soil and variations in C content at depth across this landscape, and is an important consideration when defining landscape scale DOC mobilization and redistribution dynamics, especially in wet boreal landscapes.

**4.3 DOC flux and water flux relationship varies with seasonal environmental change and suggests an interactive temperature control**

Despite the dominant water input control, the relationship between DOC flux and water flux was not consistent at the seasonal scale (Fig. 3; Table 3). Soils of both plot types appeared to be flushed during periods of high, continual leaching and low temperatures (Fig. 1), suggesting that the seasonally variable production of DOC and/or water soluble organic carbon (WSOC)

is an important secondary control. Some field studies have shown that soil DOC concentrations remain constant and do not become more dilute with increasing soil water fluxes, suggesting that the pool of WSOC is not easily exhausted in those systems (Karsten Kalbitz, Meyer, Yang, and Gerstberger, 2007; Klotzbücher et al., 2014). This leads to proportional increases in DOC flux with increasing water flux and therefore, a water limitation on DOC mobilization. While summer (Fig 3a), and likely winter, DOC fluxes in this study were similarly water-limited, autumn and spring snowmelt fluxes exhibit a tapering off

of DOC fluxes during periods of highest water input (Fig 3bc), suggesting a production limitation during autumn and snowmelt.

Since DOC flux was calculated as the product of DOC concentration and solution volume for each measurement period, the highest periods of DOC flux occur when conditions support relatively high values of both terms. This occurred most frequently during late summer/early autumn and ecologically requires the combination of: (1) the production of water-soluble organic

carbon (WSOC) or DOC via temperature sensitive mechanisms such as SOM and/or litter decomposition rhizodeposition, and microbial biomass turnover (Christ & David, 1996; Kalbitz et al., 2007; Weintraub et al., 2007), and (2) sufficient water inputs to result in a soil water flux that mobilizes or extracts DOC from O horizons. Soil water fluxes were negatively correlated with soil temperature in this study (Table 3a), likely driven by the seasonal temperature dependence of net water input and





evapotranspiration, while DOC concentration was positively correlated with soil temperature. Therefore, the seasonality of DOC flux involves an interactive temperature effect, where T dependent biogeochemical processes and T dependent soil water fluxes interact to form seasonally unique combinations or scenarios important to a predictive understanding of these fluxes.

### 4.3.2 Water Limited Scenarios: Summer and Winter

Fluxes of water and DOC were dynamic on the weekly to monthly scale during all seasons except winter (Fig 1e,f), revealing that flux conditions can occur at all times of the year in these sites, except during periods of deep, consistent snowpack which limits water input to the soil and, consequently, DOC mobilization. Summer also exhibited a water limitation on DOC mobilization but on a shorter time scale, alternating between weekly periods of no water and DOC flux and periods of large water and DOC fluxes. While we detected no relationship between DOC flux and soil moisture using the whole dataset (Table

3b), it is likely that antecedent soil moisture can affect the proportion of the water input that results in a water and DOC flux in the summer when soil drying-rewetting cycles were common (Fig. 1; grey shaded bars), although this doesn't appear to be a driving factor throughout the year in these plots.

In summer, when $CO_2$ efflux rates were high but DOC fluxes were intermittent, $CO_2$ was, in part, a larger loss of soil C because

insufficient water input limited mobilization of DOC from O horizons, and the lack of mobilization may additionally mean loss of DOC via respiration (Moore et al., 2008). In early autumn however, the elevated water flux, cooler temperatures, and decreasing $CO_2$ efflux rates, favour a larger proportion of the SOC pool being mobilized as DOC and lost to downstream C pools either in mineral soil or further to groundwater and headwaters.

### 4.3.2 Production Limited Scenarios: Autumn and Snowmelt

With continuous leaching and decreasing soil temperatures, late autumn water inputs resulted in a decrease in DOC concentrations and DOC fluxes, such that soils appear to be flushed of the WSOC or DOC pool just prior to snowpack development. Thus the availability of the extractable DOC pool in these soils during the snowpack and subsequent snowmelt period was likely much reduced by high autumn water input at low soil temperatures. Spring snowmelt captured during this study year followed a winter of constant snow cover and contributed approximately 31% of the annual water input to the soil,

and 20% of the annual DOC flux, but occurred over a period that represented only 13% of the year. Despite representing the largest hydrological event during this study year, the large water flux over a short time period combined with relatively low soil temperatures and previously flushed soils, resulted in dilute leachate (low DOC concentration) and a smaller contribution to the annual DOC flux in relation to early autumn fluxes.

### 4.4 Climate change impacts on seasonal soil conditions and precipitation patterns will affect DOC fluxes

This study shows that DOC flux variation is well described by water flux variation, but that gradual flushing of O horizons occurs during consistent leaching events throughout autumn as soil temperatures decrease. These seasonal trends suggest that





the projected increases in precipitation at mid to high latitudes in the northern hemisphere (Kirtman et al., 2013) may result in proportional increases in DOC fluxes in the summer and early autumn when soil temperatures are warm in this system, but that DOC or water-soluble OC (WSOC) pools are depleted during seasonal decreases in soil temperature. In order for increasing water fluxes to result in increased losses of DOC they must therefore be met with increased production of DOC/

WSOC; a process dependent on how increases in precipitation are seasonally distributed. One potential mechanism of increased WSOC production that is particularly relevant to this system is the increased occurrence of soil frost. No soil freezing occurred under the consistently deep snowpack conditions observed during winter in this study. With warm winter conditions expected to become more frequent in northern regions, melting and reforming of the snowpack over winter will have consequences for soil exposure and frost, as well as the frequency and magnitude of winter-time water flux events. Similar to

soil drying-rewetting events (Fierer and Schimel, 2002), soil freeze-thaw cycles have been shown to increase soil DOC concentrations by disturbing soil, root and microbial structures (Haei et al., 2013; Schimel and Clein, 1996). Increased winter rainfall events and within winter snowmelt that would drive larger winter soil water fluxes, combined with soil freeze-thaw events, could therefore contribute to future increases in wintertime losses of DOC, but decreases in spring snowmelt losses. Therefore, the effect of climate change on DOC fluxes will likely depend on the redistribution of intra-annual precipitation

and form, and the indirect effect of these changes on soil structure. In addition, this study highlights that the interactive temperature effect, as well as horizontal flow dynamics are important areas of inquiry for defining the role of DOC in the carbon budget of forest landscapes enabling prediction of the response of forest C balance to disturbances such as harvesting and climate change.

*Data availability.* All data are included in the paper tables and the Supplement.

*Supplement.* The supplement related to this article is available online at:

*Author contributions.* KAE and SEZ designed the study with input from KB. KB and KAE designed the lysimeters and planned

their installation as well as installation of all environmental monitoring equipment. KB collected and analysed the lysimeter, environmental monitoring and soil properties data. XZ contributed the soil respiration data and analysis. KB prepared the paper, with editing from SEZ and KAE and further contributions on final drafts from XZ.

*Competing interests.* The authors declare that they have no conflict of interest.


*Acknowledgements.* Special thanks for field assistance provided by individuals at the Atlantic Forestry Centre (Corner Brook) of Natural Resources Canada: Andrea Skinner, Darrell Harris, and Gordon Butt; and Memorial University, Grenfell campus: Sarah Thompson and Danny Pink, as well for laboratory assistance provided by Jamie Warren at Memorial University, St. John's campus. The throughfall carbon inputs were estimated based on collections made and analysed by Alex Newman in




2015. Funding was provided by the Centre for Forestry Science and Innovation, Agrifoods and Forestry, Government of Newfoundland and Labrador, Natural Sciences and Engineering Research Council (NSERC) Strategic Partnerships Grants (479224-15) and the Canada Research Chairs Program.

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





**Table 1.** Ecosystem and soil C properties of black spruce forest and adjacent harvested plots. Values are means of 12 litterfall traps per plot type, 16 soil respiration collars per plot type, 3 organic (O) horizon soil temperature and moisture probes per plot type, 2 mineral horizon soil temperature and moisture probes per plot type, 9 O horizon samples per plot type used to determine thickness, % C, C stock, C:N and bulk density, 1 snow pit per plot type, and 3 seasonally distinct rain collections used together with annual rainfall to estimate an annual C input, with standard error in parenthesis. Results for one way ANOVAs (litterfall, O horizon thickness, and soil %C, C stock, C:N, soil bulk density) and T-tests (soil temperature and moisture) conducted to identify plot type differences are shown where applicable with significant results in bold (alpha=0.05). Soil moisture is measured as volumetric water content (VWC). See methods for further measurement and sample collection details.

| | Forest | Harvested | T value | F value | p value |
|---|---|---|---|---|---|
| Litterfall | | | | | |
| Total mass (g m$^{-2}$ y$^{-1}$) | 240.9 (14.7) | 13.7 (3.2) | - | 309.0 | **<0.0001** |
| Total carbon (g C m$^{-2}$ y$^{-1}$) | 212.4 (14.3) | 12.4 (2.9) | - | 287.6 | **<0.0001** |
| | | | | | |
| Rain (g DOC m$^{-2}$ y$^{-1}$) | 5.5 | 3.9 | - | - | - |
| Snow (g DOC m$^{-2}$ y$^{-1}$) | 2.1 | 1.3 | - | - | - |
| | | | | | |
| Soil Respiration (g C m$^{-2}$ y$^{-1}$) | 711.9 (59.5) | 672.2 (32.3) | - | 0.226 | 0.651 |
| | | | | | |
| Organic horizon | | | | | |
| Soil T (℃) | 6.4 (0.03) | 7.6 (0.12) | -11.31 | - | **0.00291** |
| Soil M (VWC) | 13.2 (0.8) | 20.2 (5.4) | -1.289 | - | 0.321 |
| Thickness (cm) | 8.17 (0.6) | 4.26 (0.6) | - | 18.37 | **0.0128** |
| % C | 47.6 (0.7) | 43.0 (2.7) | - | 1.07 | 0.121 |
| C stock (kg C m$^{-2}$) | 2.39 (0.18) | 1.34 (0.26) | - | 12.15 | **<0.0001** |
| Soil bulk density (g/cm$^3$) | 0.06 (0.007) | 0.07 (0.004) | - | 3.08 | 0.154 |
| Mineral horizon (top 15 cm) | | | | | |
| Soil T (℃) | 6.5 (0.2) | 7.6 (0.1) | NA | NA | NA |
| Soil M (VWC) | 40.1 (2.0) | 48.3 (2.6) | NA | NA | NA |
| % C | 2.63 (0.41) | 2.17 (0.42) | - | 0.996 | 0.375 |
| C stock (kg C m$^{-2}$) | 3.85 (0.79) | 5.33 (0.81) | - | 3.123 | 0.152 |
| Soil bulk density (g/cm$^3$) | 2.3 (0.4) | 2.8 (1.3) | - | 0.121 | 0.746 |
| % rock by volume | 84 (3) | 64 (7) | - | 0.355 | 0.133 |





**Table 2.** Pearson correlations between lysimeter captured dissolved organic carbon concentrations (mg DOC L$^{-1}$), dissolved organic carbon fluxes (g DOC m$^{-2}$ d$^{-1}$), soil solution fluxes (L water m$^{-2}$ d$^{-1}$ ) and environmental variables (mean soil temperature,  mean soil moisture and daily water input rain and/or snowmelt) over 30 collection periods.

| | df | A. mean soil temperature (℃) | | B. mean soil moisture (VWC) | | C. total water input (L m$^{-2}$ d$^{-1}$) | |
|---|---|---|---|---|---|---|---|
| | | F | H | F | H | F | H |
| mg DOC L$^{-1}$ | 23 | r= 0.9493 | r= 0.8083 | r= -0.2383 | r= -0.4773 | r= -0.4325 | r= -0.5431 |
| | | t= 7.7154 | t= 6.5847 | t= -1.1770 | t= -2.6052 | t= -2.3008 | t= -3.1022 |
| | | **p< 0.0001** | **p<0.0001** | p= 0.2512 | **p= 0.0158** | **p= 0.0308** | **p= 0.0050** |
| g DOC m$^{-2}$ d$^{-1}$ | 28 | r= -0.1387 | r= -0.1575 | r= -0.1282 | r= -0.1454 | r= 0.7358 | r= 0.6113 |
| | | t= -0.7412 | t= -0.8437 | t= -0.6843 | t= -0.7779 | t= 5.7500 | t= 4.0880 |
| | | p= 0.4647 | p= 0.4060 | p= 0.4994 | p= 0.4431 | **p<0.0001** | **p= 0.0003** |
| L water m$^{-2}$ d$^{-1}$ | 28 | r= -0.5383 | r= -0.5683 | r= 0.0252 | r= -0.0602 | r= 0.8142 | r= 0.8810 |
| | | t= -3.3799 | t= -3.6550 | t= 0.1336 | t= -0.3190 | t= 7.4214 | t= 9.8511 |
| | | **p= 0.0021** | **p= 0.0011** | p= 0.8947 | p= 0.7521 | **p<0.0001** | **p<0.0001** |





**Table 3**. Two-way ANOVA results examining the effect of water input, season and the interaction on DOC fluxes. Data plotted in Figure 3

| DOC Flux | df | F value | p-value |
|---|---|---|---|
| Water Input | 1 | 79.1618 | **<0.0001** |
| Season | 2 | 11.3778 | **<0.0001** |
| Water Input x Season | 2 | 5.4857 | **0.0067** |



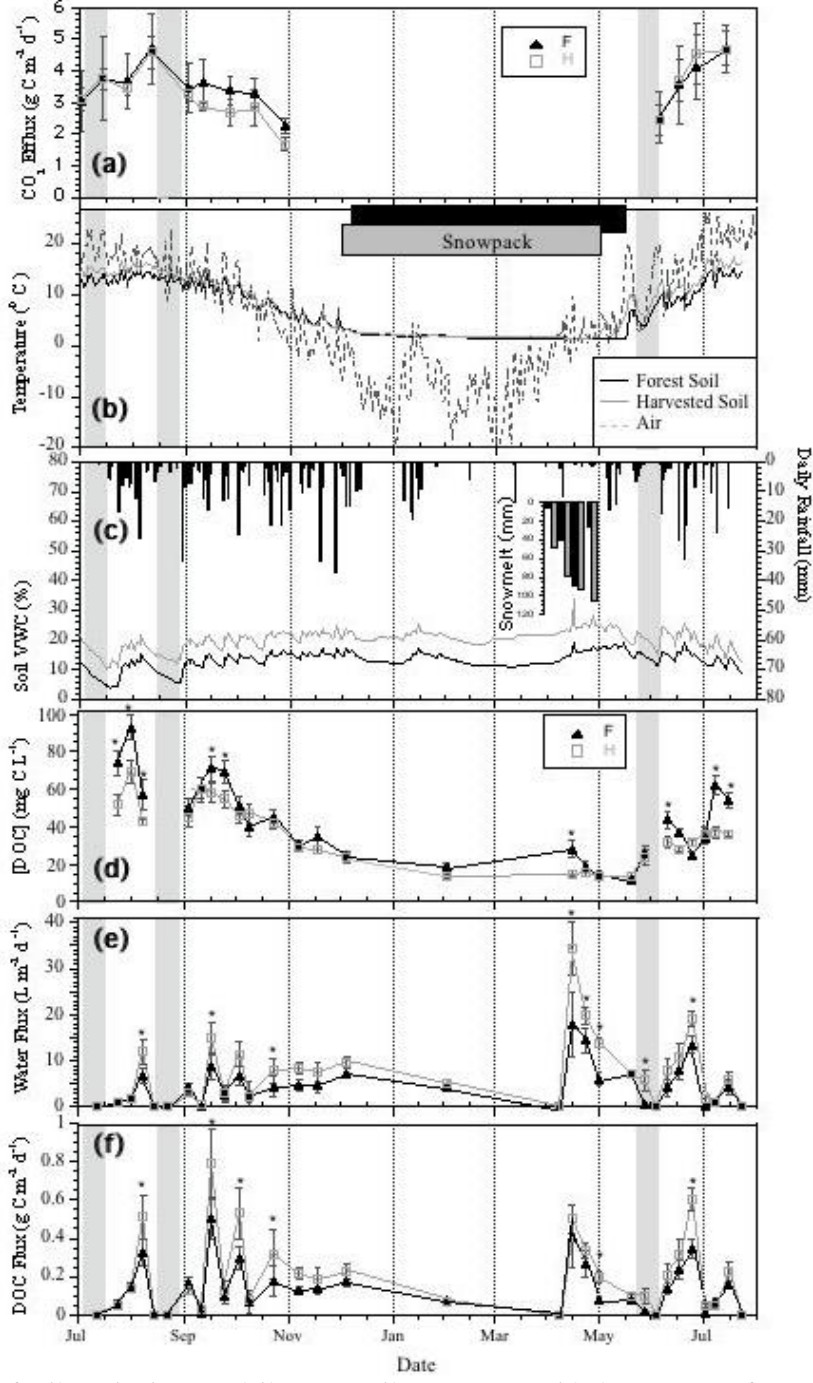

**Figure 1.** Temporal variation of soil respiration (a), daily mean soil temperature with the presence of a snowpack indicated by the grey (harvested) and black (forest) bar (b), daily rainfall and daily mean soil moisture (c), and lysimeter collections (d,e,f) from July 2013 to July 2014 in black spruce forest and harvested plots. Snowmelt was estimated using measured changes in snow depth in forest (black bars) and harvested (grey bars) plots. The mean dissolved organic carbon (DOC) concentration (d), water flux (e), and DOC flux (f) was determined using passive pan lysimeter collections underneath O horizons. Lysimeter sampling was continuous and points represent a mean daily flux over each collection period. Error bars show standard error of the mean of 12 lysimeter collections per plot type per collection period. Grey shading areas indicate dry periods signified by those exceeding 10 consecutive days of rainfall less than 10mm/day, corresponding to periods of soil drying. Significant differences in DOC flux, water flux and DOC concentration between plot type on each collection day where determined by repeated measure linear mixed model post hoc tests and are indicated by an asterisk (alpha = 0.05).





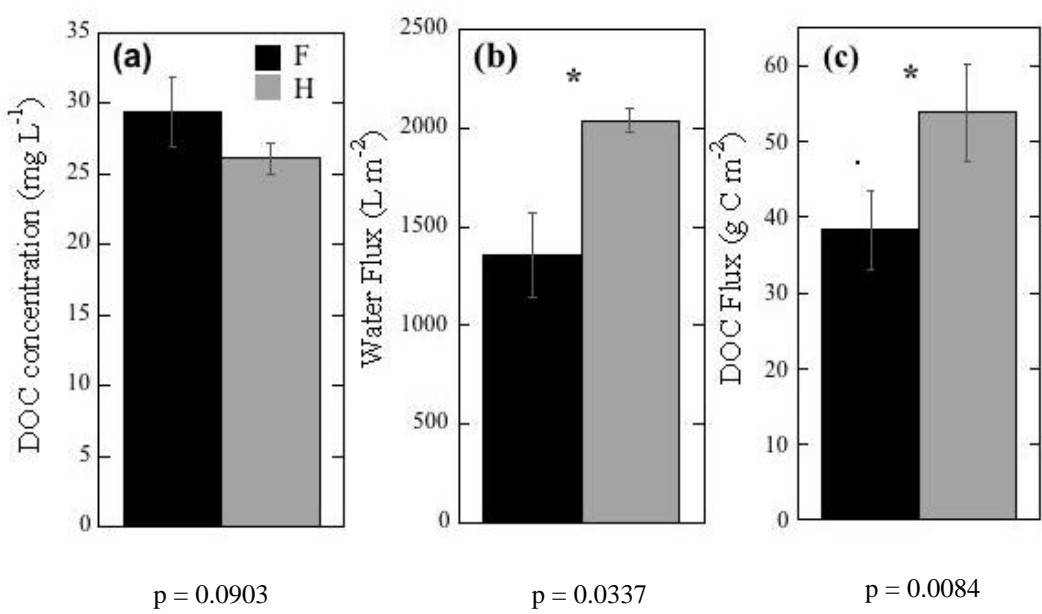

**Figure 2**. Mean annual lysimeter collected variables. Volume weighted dissolved organic carbon (DOC) concentration (A), total water flux (B), and total DOC flux collected from organic horizons of forest (F) and harvested (H) plots over the entire study period. Annual values were calculated from the accumulated 29 sample collection time points taken from 12 F and 12 H passive pan lysimeters over one year from July 2013 to July 2014. Asterisks show significant differences between plot type (alpha = 0.05) determined using one-way plot nested ANOVA tests (Table S2).





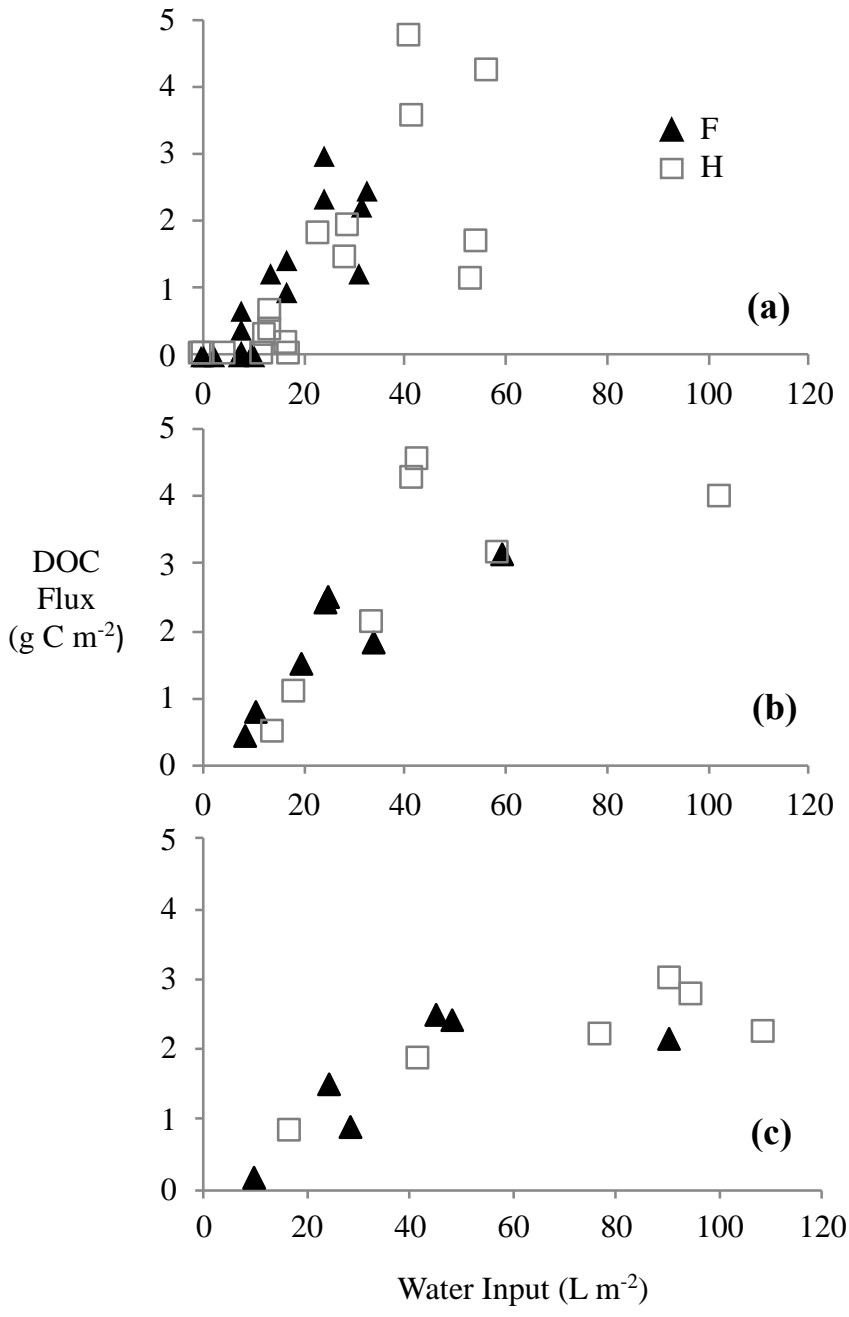

**Figure 3.** Seasonal relationship between dissolved organic carbon (DOC) fluxes and water input to the soil in mature forest (F) and harvested (H) plots. Seasons are designated as summer (a), autumn (b) and winter/snowmelt (c).