# Peer review of "Dissolved organic carbon mobilized from organic horizons of mature and harvested black spruce plots in a mesic boreal region"

_Biogeosciences, 2018_

## Referee Comment (RC1) · L. Thieme (Referee) · 11 Mar 2019

This manuscript by Bowering et al. presents a study of seasonal variations on concentrations and fluxes of DOC as well as soil respiration from organic horizons of mature boreal spruce forests and of harvested sites 10 years after clear cutting.

The authors provided a lot of details regarding their sampling schemes, their approaches and their analysis. Very well done. The overall presentation is well structured and clear. The manuscript contains a bulk of detailed information about how DOC concentrations and fluxes, soil temperature and moisture and soil respiration changes along the year and highlights statistical differences between the two plot types (mature

and harvested). Their discussion on organic horizon DOC flux dynamics and relation to water fluxes as well as the possible impact of climate changes are sound.

One minor comment regarding Figure 1: The axis captions are hard to read. The same applies to the error bars and the asterisks in panel d-f.

---

## Referee Comment (RC2) · Anonymous Referee #2 · 6 Apr 2019

The manuscript presents a thorough assessment of DOC fluxes in boreal landscapes and how they might be affect by forest harvesting and climate change. Principally, the study is well designed and the manuscript is nicely written and the results contribute to our understanding of DOM mobilization in boreal forests. The major shortcoming of the manuscript is the estimate of water (and thus DOC) fluxes that is based on water collected using passive pan lysimeters. Although they seemed to be well designed (using glass beads to mimic a hydrological continuum), it remains uncertain how well they functioned (e.g. by tracer). While water recovery was tested to be 90%, measured water drainage was found to exceed rainfall inputs (+50%) and thus measured drainage was about twice as high as what one could expect. In the discussion, the discrepancy

was explained by lateral flow contributing. This implies that the lysimeters acted as funnels draining a greater footprint area and thus, comparisons of DOC fluxes with soil CO2 effluxes are not valid as they originate from different areas. To me an appropriate estimate of water fluxes seems crucial for the manuscript as the discussion centers all around a mass balance comparing DOC with soil CO2 effluxes. I would strongly recommend to use a water balance model to estimate DOC export from the organic layer or provide clear evidence on lateral flow or the footprint area. More information on the set-up of the lysimeters and the site conditions (slope) should be added. In contrast to the uncertainties related to the quantitative estimates, conclusions made in relative terms e.g. harvest effects, seasonality etc. are still valid and merit publication.

Specific comments:

Abstracts L. 23 ff An Abstract should be informative and contain the key data. The implication/conclusion section is much too long, 10 lines. I missed values and comparison with soil CO2 effluxes and forest management aspects.

Methods Page 5, Line 5ff lysimeter set-up "It was desirable for this study"...please describe what was exactly done and give details on glass beads (size classes), depths of the glass bead layer, length x width of the lysimeter, connection of lysimeter to sample container etc.. How was it installed? Was the organic layer completely removed beforehand? A sketch added to the Supplemental Information might be helpful. According to the test described it seems that lysimeters functioned well but why did they not collect lateral water in your test but later during the regular monitoring? The appropriate capturing/estimate of water fluxes is crucial for estimating DOC fluxes and thus lysimeters known to create sampling artefacts should be tested rigorously (e.g. by a tracer) or backed up with modelling of water fluxes.

Page 8, Line 15 453 cm as snowfall, typo? If indeed snow depth is meant, please transform it to water equivalent.

Page 8, line 19 I would recommend to report no decimal for rainfall (which is beyond

any precision possible)...

Page 9, Line 26 clarify that you mean the SOC stock in the organic layer.

Page 10 How can the water flux in the O horizon (1366 and 2040 mm) exceed or be in the same range as the input via rainfall (1305mm)? Estimates of water fluxes are crucial as DOC fluxes directly depend upon water fluxes. Generally, this is done via modelling of water fluxes (see papers by Fröberg et al., Kindler et al., 2010 GCB). The values you provide indicate that the lysimeters worked well (which is not always the case) but that they might fetch water from a greater area or include a lateral component. How does the topography of the site looks like (no information given in the methods...).

Page 10 Line 16 please rephrase the sentence – and clarify that 'corresponding to a total depth of 84 cm and 110 cm' was the snow depth when snow/water was sampled (?)

Discussion Page 11, Line 13ff As the DOC fluxes seem to be very high due to an overestimate of water fluxes, the discussion includes a high uncertainty. At a rainfall of 1300 mm, evaporation rates of 100-200 mm and a evapotranspiration of approx. 3-500 mm, the DOC fluxes are probably a factor of two smaller than estimated here. This is also relevant for the comparison with other C fluxes/pools.

Page 11, Line 30ff here it needs to be clarified that the greater water flux drives the management effects

Page 12, Discussion of lateral water fluxes. The appropriate estimation of water fluxes is crucial for the overall manuscript (and appears very late in the discussion. Based on the values given, I was wondering much earlier that something went wrong). Lysimeters are known to have artefacts as they alter the soil continuum: they can act either as a funnel or as a barrier depending on the soil conditions. I would thus not rely on the assumption that the lysimeters used here captured water fluxes (horizontal and lateral ones) correctly. Probably, there is lateral flow (what is the slope of your site?),

but the estimate provided here is too speculative. Moreover, is laterally moved DOC a real export? How can you compare total DOC export (lateral and vertical) with soil CO2 effluxes in quantitative terms? I recommend to model water fluxes and use these values to estimate vertical DOC loss from the O-horizon.

Page 14 comparison with soil CO2 effluxes. You might estimate the seasonal pattern of DOC vs. soil CO2 effluxes (or their temperature dependencies. DOC production was found to be less temperature dependent than CO2 production (in soil warming studies). Table 1 : Mineral soil bulk density of 2.8 g/cm3 is hardly possible as rock density is generally assumed to be 2.65 g/cm3

---

## Referee Comment (RC3) · Anonymous Referee #3 · 30 Apr 2019

The study by Bowering and coauthors presents a thorough survey of carbon exchanges between above and belowground terrestrial pools, compared across pristine and harvested boreal Canadian landscapes. The findings are linked to environmental conditions, in the context of changing climate and hydrology in the region. The relatively high temporal resolution of the dataset provides insight into cross-season differences in the controls on soil DOC export between harvested and pristine plots, a clear strength of the paper. Also, I really liked how the authors explicitly discuss the importance of their findings for the parameterization of larger forest carbon cycle modelling efforts. I recommend below a few general and specific changes to the current manuscript that could further strengthen the paper.

[Figure]

General:

-Introduction as written does not cover the effects of forest harvesting and the state of knowledge regarding forest/soil C cycling impacts. A bit of context here is important because the cross comparison of plot types is a big theme. Also, page 11, lines 22 and on contains key findings that would be better showcased if the effects/unknowns related to harvesting are introduced earlier in the paper. To that end I recommended below citing a recent review on this topic (James & Harrison. Forests 2016, 7, 308; doi:10.3390/f7120308) that could be used as context in the introduction and discussion.

-Consider adding a simple drawing that summarizes the fluxes and pools of C measured here, perhaps boxes and arrows sized to pool sizes and flux rates, respectively. Not critical since table 1 has much of the information, but a figure like this could really help readers follow key findings as they are presented in the discussion.

-The concept of the net ecosystem carbon budget (NECB; Chapin et al. 2006 Ecosystems; Webb et al. 2018 Ecosystems for a nice review) is not directly presented, but could be useful context. Even though not every single C flux is measured here, the discussion does revolve around this concept, and the authors are measuring a key flux term (hydrologic DOC export) that has often been overlooked in earlier efforts to build C budgets. Consider introducing this early in the introduction and again in the first 2 paragraphs of the discussion. Such discussion would fit nicely with the summary drawing figure suggested above.

Specific:

P1, l. 18-26. Abstract could be shortened. Consider summarizing results/correlations more succinctly.

P1,l.25. Flushing means what exactly? DOC removal? Maybe say flushing of DOC.

P3,l.16. grammar

P3,l.20. Could add conclusion sentence summarizing the outstanding issue that is

motivating your study.

P7,l.4. Add shot sentence explaining how soil respiration calculated.

P7,l.20. What package in R was used for the LMEs?

P8,l.26. Introduce the soil thickness measurements shown in Table 1 here too.

P8,l.26. In Fig. 1, reorder the panels so that soil respiration is numbered according to when it is introduced.

P8,l.31. What do you mean by partial melt?

P9,l.7. Should current fig. 1b be current fig. 1c?

P10,l.3. reword "were not found" to "was" if singular.

P10. Order of figure introduction is confusing throughout entire page. Could rearrange existing text so that corresponding panels from Fig. 1 introduced first, Fig. 2 second.

P10,l.11. How much? Consider adding a percentage value.

P10,l.16. Snow depth?

P10,l.18. Rain throughfall?

P11,l.9. Add "was" before "observed".

P13,l.12. Take pgph 1 step further with conclusion sentence that links back to your results.

P13,l.24. Whys is winter included here? Don't 3a and 3b depict linear increases, while 3c depicts the plateau? Should the reference to fig. 3b be included in line 25? Maybe I missed something but this could be clarified.

P14,l.1-3. Excellent conclusion. Consider repeating exactly like this in the abstract to shorten there.

P14,l.5. Tough to support the statement that winter fluxes were "dynamic" with only 1 measurement there, so consider rewording that.

P14,l.18. Could end this section with stronger discussion of the implications of these results. Same comment goes for the next section too. Is the timing of the precipitation the key? How well is this established in earlier studies? Could take this back to the broader literature.

P15, l.5. Important end to the sentence, but awkward as currently written. Consider rewording.

Fig. 1. Center the Y-axis titles on each panel.

Fig. 3. Consider adding trendlines to quantify the different seasonal relationships.

---

## Author Response (AR1)

Response to interactive comments of Reviewer 1 (bg-2018-516)

We thank L. Thieme for the helpful comments. Our response to specific comments (reprinted in bold) are provided below

**This manuscript by Bowering et al. presents a study of seasonal variations on concentrations and fluxes of DOC as well as soil respiration from organic horizons of mature boreal spruce forests and of harvested sites 10 years after clear cutting.**
**The authors provided a lot of details regarding their sampling schemes, their approaches and their analysis. Very well done. The overall presentation is well structured and clear. The manuscript contains a bulk of detailed information about how DOC concentrations and fluxes, soil temperature and moisture and soil respiration changes along the year and highlights statistical differences between the two plot types (mature**
**and harvested). Their discussion on organic horizon DOC flux dynamics and relation to water fluxes as well as the possible impact of climate changes are sound.**
**One minor comment regarding Figure 1: The axis captions are hard to read. The same applies to the error bars and the asterisks in panel d-f.**

Will revise axis captions, error bars and asterisk so that they are easier to read. Done

Response to interactive comments of Reviewer 2 (bg-2018-516)

We thank reviewer 2 for the helpful comments that will aid in significantly improving our paper, particularly through clarification of the measurements made and our interpretations of water and DOC mobilization in the boreal watershed context. Our response to specific comments (reprinted in bold) are provided below

**The manuscript presents a thorough assessment of DOC fluxes in boreal landscapes and how they might be affect by forest harvesting and climate change. Principally, the study is well designed and the manuscript is nicely written and the results contribute to our understanding of DOM mobilization in boreal forests. The major shortcoming of the manuscript is the estimate of water (and thus DOC) fluxes that is based on water collected using passive pan lysimeters. Although they seemed to be well designed (using glass beads to mimic a hydrological continuum), it remains uncertain how well they functioned (e.g. by tracer). While water recovery was tested to be 90%, measured water drainage was found to exceed rainfall inputs (+50%) and thus measured drainage was about twice as high as what one could expect. In the discussion, the discrepancy was explained by lateral flow contributing. This implies that the lysimeters acted as funnels draining a greater footprint area and thus, comparisons of DOC fluxes with soil CO2 effluxes are not valid as they originate from different areas. To me an appropriate estimate of water fluxes seems crucial for the manuscript as the discussion centers all around a mass balance comparing DOC with soil CO2 effluxes. I would strongly recommend to use a water balance model to estimate DOC export from the organic layer or provide clear evidence on lateral flow or the footprint area. More information on the set-up of the lysimeters and the site conditions (slope) should be added. In contrast to the uncertainties related to the quantitative estimates, conclusions made in relative terms e.g. harvest effects, seasonality etc. are still valid and merit publication.**

1) Many studies have investigated DOC fluxes as predominately vertical transfers of carbon from organic horizons to the lower mineral soil. The motivation for this study was to understand and discuss DOC dynamics within a hillslope (5-13 % gradient range across plots) to aid in understanding DOC flux dynamics at the watershed scale which has not been well documented. The conceptual idea:

   Precipitation that infiltrates the soil surface flows both vertically and horizontally depending on landscape slope, the relative permeability of soil and vegetation layers, antecedent soil moisture and lack or presence of a snowpack. Overlying the mineral soil are 2 layers of permeable material (the organic horizon and moss layer). In winter, the snow also serves as a permeable layer. Therefore, lysimeters potentially collected water that infiltrated vertically through the snow and/or moss and organic horizons, along with additional water that moved laterally through those layers into the lysimeters from upslope. These flow paths are seasonally dependent.

The 90% lysimeter efficiency result did not unfortunately test appropriately for this phenomenon as lysimeters were only watered directly above the dimensions of the catcher to determine if the design was working (i.e. plumbing all connected between the pan and downslope, buried collection carboy). We do not know what the actual total footprint beyond the lysimeter dimensions is, and therefore that value is not a true description of the lysimeters ability to capture rainfall.  We will clarify the purpose of our test and the appropriate application in the methods. A tracer test may have helped us estimate how much of the water flux measured here was from lateral flow upslope, although a better approach will be through complementary use of a model. See our proposal below in item 2.

2) While we acknowledge that lysimeters alter soil hydraulic properties making accurate quantification of water fluxes difficult, modelling of water flow also has limitations especially at the organic-mineral horizon interface measured in this study. We will run a model of water flow in order to better facilitate discussion of the two approaches and their respective limitations. We've assessed the requirements of the COUPModel, and have confirmed with the creator of the model that we have the necessary parameters to run this exercise as a supplement to our measurements. Incorporating such a modelling approaches should provide evidence for the relative magnitude of lateral flow and constrain the water fluxes measured. This will enable us to more accurately discuss the hillslope DOC fluxes in the watershed context where both vertical and horizontal flow are relevant.

3) The manuscript discussion was not meant to be centered around a mass balance of DOC with soil $CO_2$ effluxes. We can see how this could be misinterpreted given the title of the first discussion heading and following paragraph.  Both values were included as a means of comparing the magnitude of those two ecosystem C fluxes in this boreal system, demonstrating that although DOC fluxes are small in comparison to soil $CO_2$ effluxes they are similar in magnitude to boreal NEP estimates. Losses of DOC from the ecosystem could potentially affect NEP especially in the harvested stands where water fluxes remain elevated. Further work should be done to investigate the extent of this effect as our manuscript only offers that information as an observation and not as a key finding.

The discussion was reorganized in the revised manuscript to place less emphasis on a mass balance comparison, instead highlighting the more impactful results regarding effects of harvesting and seasonality of DOC.

We investigated the COUP model and found that modeling of water fluxes is conducted based on soil texture and hydraulic properties of mineral soils together with the Richard's equation. However, macropore water flow can generate rapid lateral subsurface water flow (Beven and Germann, 2013, Water Resources Research), especially in highly porous organic soils that sit above mineral soils of much lower soil hydraulic conductivity. In order to assess the impact of macropore flow on our lysimeter

collections we conducted a series of infiltration experiments (see Page 3: lines 22 - 32 and Page 12: lines 5-26; as well as the addition of Table 4) that determined the water content of O horizons at residual, matrix and macropore saturation. We used these values to calibrate and assess our continuous field measurements of soil water content in O horizons, which helped us to determine during which lysimeter collection periods lateral flow was likely to have occurred (see added Figure 5). Lysimeters collected water in excess of water input when matrix saturation, or the initiation of macropore flow, had been reached. This exercise highlighted O horizon soil hydrology modelling as an important area for further investigation. The O horizons are a key sources of DOC and accurate modelling of water is necessary for defining the role of DOC at the watershed and ecosystem scales. This cannot be done based on methods that ignore macropore water flow dynamics.

**Specific comments:**
**Abstracts L. 23 ff An Abstract should be informative and contain the key data. The implication/conclusion section is much too long, 10 lines. I missed values and comparison with soil CO2 effluxes and forest management aspects.**

The abstract will be shortened with greater emphasis on the key findings rather than implications and conclusions. Done

**Methods Page 5, Line 5ff lysimeter set-up "It was desirable for this study". . .please describe what was exactly done and give details on glass beads (size classes), depths of the glass bead layer, length x width of the lysimeter, connection of lysimeter to sample container etc.. How was it installed? Was the organic layer completely removed before- hand? A sketch added to the Supplemental Information might be helpful. According to the test described it seems that lysimeters functioned well but why did they not collect lateral water in your test but later during the regular monitoring? The appropriate capturing/estimate of water fluxes is crucial for estimating DOC fluxes and thus lysimeters known to create sampling artefacts should be tested rigorously (e.g. by a tracer) or backed up with modelling of water fluxes.**

We will add a sketch to provide more details regarding the design of the lysimeters used in this study and with that include more details on the steps taken to install these lysimeters. Done

The test conducted only entailed water applied to the actual lysimeter footprint, which will be described in the added methodological details, and not any upslope or downslope areas around that footprint. We recognize that this was not ideal as it did not assess lateral flow. However, by incorporating the modelling comparison as suggested and described above we should be able to place some constraints on what the lysimeter water fluxes provide.

The COUP model was assessed but found to not be representative of our system. Infiltration experiments conducted on O horizons helped us to identify periods of matrix saturation and macropore-driven lateral flow.

**Page 8, Line 15 453 cm as snowfall, typo? If indeed snow depth is meant, please transform it to water equivalent.**

"453 cm as snowfall" should read 453 mm water equivalents as snowfall. This will be changed in the revised manuscript. Done

**Page 8, line 19 I would recommend to report no decimal for rainfall (which is beyond any precision possible). . .**

Will be revised Done

**Page 9, Line 26 clarify that you mean the SOC stock in the organic layer.**

Will be revised Done

**Page 10 How can the water flux in the O horizon (1366 and 2040 mm) exceed or be in the same range as the input via rainfall (1305mm)? Estimates of water fluxes are crucial as DOC fluxes directly depend upon water fluxes. Generally, this is done via modelling of water fluxes (see papers by Fröberg et al., Kindler et al., 2010 GCB). The values you provide indicate that the lysimeters worked well (which is not always the case) but that they might fetch water from a greater area or include a lateral component. How does the topography of the site looks like (no information given in the methods. . .).**

We will include more information regarding the topography of the site in the methods. Done

Regional as well as plot level data was used in the estimates of water input via precipitation (rain + snow). The 1305 total annual precipitation recorded could be an underestimate of this input at the plots especially because snowfall was not measured at the plot level but was used from a meteorological station 50km away from the site in Deer Lake, NL.  The on-site snowpack data we have available prior to snowmelt (84 and 110 cm in the forested and harvested stands) was deeper than the maximum snow on ground measured at the Deer Lake weather station further suggesting an underestimate of water input as snow at the site level.  We will provide more detail and constraints on the estimate of water input to these plots in the revision.

Secondly, yes, lateral flow in this system is very likely given differences in permeability between surface layers (snow, moss and organic layers) and deeper mineral soil layers, as well as the slope of the landscape (5-13%). It is certainly possible, therefore, for soil water fluxes to exceed input via precipitation.  In fact our headwater catchment hydrology indicates a good match between discharge and lysimeter water fluxes during snowmelt, a period of little to no evapotranspiration.  It is, however, difficult to determine how much of the exceeding soil water flux is driven by natural lateral flow and how much is an artefact of the lysimeter.  This is where comparison to a model could be beneficial, although models of water flow also have their limitations. Both approaches  are necessary to come closer to a real world description. An

exclusive vertical flow application undervalues the data presented, therefore, we will assess vertical and horizontal flow model results for this site.

Done, see explanation above.

**Page 10 Line 16 please rephrase the sentence – and clarify that 'corresponding to a total depth of 84 cm and 110 cm' was the snow depth when snow/water was sampled (?)**

Yes, "84 cm and 110 cm" was the snow depth when snow was sampled. Will be revised. Done

**Discussion Page 11, Line 13ff As the DOC fluxes seem to be very high due to an overestimate of water fluxes, the discussion includes a high uncertainty. At a rainfall of 1300 mm, evaporation rates of 100-200 mm and a evapotranspiration of approx. 3-500 mm, the DOC fluxes are probably a factor of two smaller than estimated here. This is also relevant for the comparison with other C fluxes/pools.**

**Page 11, Line 30ff here it needs to be clarified that the greater water flux drives the management effects**

Will be clarified in the revised manuscript as per approach described above using the modelling comparison. Done

**Page 12, Discussion of lateral water fluxes. The appropriate estimation of water fluxes is crucial for the overall manuscript (and appears very late in the discussion. Based on the values given, I was wondering much earlier that something went wrong). Lysimeters are known to have artefacts as they alter the soil continuum: they can act either as a funnel or as a barrier depending on the soil conditions. I would thus not rely on the assumption that the lysimeters used here captured water fluxes (horizontal and lateral ones) correctly. Probably, there is lateral flow (what is the slope of your site?), but the estimate provided here is too speculative. Moreover, is laterally moved DOC a real export? How can you compare total DOC export (lateral and vertical) with soil CO2 effluxes in quantitative terms? I recommend to model water fluxes and use these values to estimate vertical DOC loss from the O-horizon.**

1) The range of slopes measured across plots was 5-13%.
2) Export depends on the area of interest. It certainly could be a real export if DOC is leaving from a fixed area, even if it is a source to downslope O horizons. We will be sure to revise in order to clarify this perspective in the context of our study site.
3) We agree that a quantitative comparison of $CO_2$ effluxes and lateral + vertical DOC fluxes is difficult and not appropriate as a mass balance approach. However, a comparison of relative quantities and dynamics is useful to demonstrate for the discussion of the relevancy of the DOC fluxes in the context of NEP.
4) We would like to maintain a position that DOC fluxes are not just vertical fluxes of C, which is an important part of understanding the role and behaviour of DOC in the

watershed context. However, we do acknowledge that accurate quantification of horizontal flow using the data currently available is not possible.

**Page 14 comparison with soil CO2 effluxes. You might estimate the seasonal pattern of DOC vs. soil CO2 effluxes (or their temperature dependencies. DOC production was found to be less temperature dependent than CO2 production (in soil warming studies).**

Will consider and revise where appropriate.

**Table 1 : Mineral soil bulk density of 2.8 g/cm3 is hardly possible as rock density is generally assumed to be 2.65 g/cm3**

We recognize the issue of the high value which is indeed elevated relative to others we have for other sites regionally and are looking into it so that we are able to correct or clarify in a revision. Corrected

Response to interactive comments of Reviewer 3 (bg-2018-516)

We thank reviewer 3 for the helpful comments that will aid in significantly improving our paper, particularly through clarification of the measurements made and our interpretations of water and DOC mobilization in the boreal watershed context. Our response to specific comments (reprinted in bold) are provided below

**The study by Bowering and coauthors presents a thorough survey of carbon exchanges between above and belowground terrestrial pools, compared across pristine and harvested boreal Canadian landscapes. The findings are linked to environmental conditions, in the context of changing climate and hydrology in the region. The relatively high temporal resolution of the dataset provides insight into cross-season differences in the controls on soil DOC export between harvested and pristine plots, a clear strength of the paper. Also, I really liked how the authors explicitly discuss the importance of their findings for the parameterization of larger forest carbon cycle modelling efforts. I recommend below a few general and specific changes to the current manuscript that could further strengthen the paper.**

 **General:**
**-Introduction as written does not cover the effects of forest harvesting and the state of knowledge regarding forest/soil C cycling impacts. A bit of context here is important because the cross comparison of plot types is a big theme. Also, page 11, lines 22 and on contains key findings that would be better showcased if the effects/unknowns related to harvesting are introduced earlier in the paper. To that end I recommended below citing a recent review on this topic (James & Harrison. Forests 2016, 7, 308; doi:10.3390/f7120308) that could be used as context in the introduction and discussion.**

The introduction will be revised to include more information on the known/unknown effects of harvesting on soils and DOC. Done

**-Consider adding a simple drawing that summarizes the fluxes and pools of C measured here, perhaps boxes and arrows sized to pool sizes and flux rates, respectively. Not critical since table 1 has much of the information, but a figure like this could really help readers follow key findings as they are presented in the discussion.**

Will consider for the revised manuscript to see if inclusion of such a figure will aid in communicating the findings more readily.  We would like to see if this can be done without creating any misinterpretations given that not all C fluxes were assessed in this study.

We have included a figure of the site and lysimeter installations (see Figure 1). The format presented avoids focus on a mass balance of the C fluxes measured (a concern of Reviewer 2), instead highlighting the DOC fluxes and the lateral and vertical fluxes that the lysimeters capture.

**-The concept of the net ecosystem carbon budget (NECB; Chapin et al. 2006 Ecosystems; Webb et al. 2018 Ecosystems for a nice review) is not directly presented, but could be useful context. Even though not every single C flux is measured here, the discussion does revolve around this concept, and the authors are measuring a key flux term (hydrologic DOC export) that has often been overlooked in earlier efforts to build C budgets. Consider introducing this early in the introduction and again in the first 2 paragraphs of the discussion. Such discussion would fit nicely with the summary drawing figure suggested above.**

We will include NECB within the first paragraph of the introduction where the fate of soil C is discussed. Done

**Specific:**
**P1, l. 18-26. Abstract could be shortened. Consider summarizing results/correlations more succinctly.**

The abstract will be shortened and edited to highlight key findings. Done

**P1,l.25. Flushing means what exactly? DOC removal? Maybe say flushing of DOC. P3,l.16. grammar**

Yes, flushing refers to the removal of DOC from soil pores during large water flux events. We will change to "flushing of DOC" to clarify. Done

**P3,l.20. Could add conclusion sentence summarizing the outstanding issue that is motivating your study.**

We will consider this idea and revise the conclusion statements accordingly. Done

**P7,l.4. Add shot sentence explaining how soil respiration calculated.**

Will add Done

**P7,l.20. What package in R was used for the LMEs?**

The "nlme" package was used. Will provide in manuscript Done

**P8,l.26. Introduce the soil thickness measurements shown in Table 1 here too.**

Will be revised Done

**P8,l.26. In Fig. 1, reorder the panels so that soil respiration is numbered according to when it is introduced.**

Will revise accordingly.Done

**P8,l.31. What do you mean by partial melt?**

Only a portion of the snowpack melted during this period. Will be revised in manuscript to clarify. Removed

**P9,l.7. Should current fig. 1b be current fig. 1c?**

Should be both 1B and 1C. Will revise Changed to fig. 1C

**P10,l.3. reword "were not found" to "was" if singular.**

Will revise Done

**P10. Order of figure introduction is confusing throughout entire page. Could rearrange existing text so that corresponding panels from Fig. 1 introduced first, Fig. 2 second.**

Will consider and revise. Revised to accommodate this suggestion where appropriate.

**P10,l.11. How much? Consider adding a percentage value.**

Will revise

**P10,l.16. Snow depth?**

Yes, will revise Done

**P10,l.18. Rain throughfall?**

Yes, will clarify Done

**P11,l.9. Add "was" before "observed".**

Will revise Done

**P13,l.12. Take pgph 1 step further with conclusion sentence that links back to your results.**

Will consider and revise Done

**P13,l.24. Whys is winter included here? Don't 3a and 3b depict linear increases, while 3c depicts the plateau? Should the reference to fig. 3b be included in line 25? Maybe I missed something but this could be clarified.**

It seems that the plateau begins towards the end of autumn when large water fluxes begin to occur, as a result of reduced ET, and that's why 3b was included. The trendlines to Fig 3 you suggest below could quantify and clarify this section. Done

**P14,l.1-3. Excellent conclusion. Consider repeating exactly like this in the abstract to shorten there.**

Great, thank you. This is helpful feedback. Will include in the abstract.

**P14,l.5. Tough to support the statement that winter fluxes were "dynamic" with only 1 measurement there, so consider rewording that.**

Will consider and reword

**P14,l.18. Could end this section with stronger discussion of the implications of these results. Same comment goes for the next section too. Is the timing of the precipitation the key? How well is this established in earlier studies? Could take this back to the broader literature.**

These are good points and would strengthen the discussion of our key results concerning seasonality of DOC. Will carefully consider how to include a stronger discussion of these implications and placing this within a broader context. Done

**P15, l.5. Important end to the sentence, but awkward as currently written. Consider rewording.**

Will revise Done

**Fig. 1. Center the Y-axis titles on each panel.**

Will revise Done

**Fig. 3. Consider adding trendlines to quantify the different seasonal relationships.**

Will consider and revise Done

[revised manuscript text omitted]

---

## Author Response (AR2)

Thank you to the reviewer for the helpful comments as they have lead to a much clearer and accurate discussion of our results. The major changes to the manuscript include an improved estimate and description of the estimate of precipitation input using corrected on site data (now discussed in a new methods section, "water input estimate"). The first section of the discussion has been revised in two major ways; (1) Omission of any comparison of our DOC flux estimates to other vertical ecosystem fluxes to better clarify that lateral fluxes contribute DOC from an area greater than 1m2, (2) clear acknowledgement that there are water and solute transport models that do incorporate macropore flow dynamics. These models, however useful, are based on mineral soil properties and focus on mineral soil hydrology. We have not found references where O horizon hydrology is explicitly considered or defined in water or solute transport models. It is our suggestion that measurement and incorporation of the specific hydrologic properties of O horizons in models would refine the landscape level role of these layers as they are both hydrologically distinct from mineral soils and are major sources of DOC in boreal landscapes.

Specific responses to comments are included below.

The authors have nicely revised their manuscript and have better clarified their sampling scheme. The results clearly shows that forest harvesting has a long-term impact on DOC fluxes and that this influence primarily occurs through changes in hydrology. Overall, this is a solid work and provides important knowledge, but the presentation and discussion of water and DOC fluxes is still too uncritical.

The main critiques are as follows:

**1. Water-DOC fluxes**.
While the authors convincingly discuss the importance of macropore and/or lateral flow by stating that water fluxes in the O horizon exceeded precipitation, it has to be clearly stated that this quantitative relationship applies only for their lysimeters used. At the landscape scale, water fluxes in the O horizon cannot exceed those of precipitation.

a. Consequently, the authors have either clarify that precipitation data used might be an underestimate as they have been measured 50 km away (which is reasonable) or clearly state that the fluxes measured here only applies for the water collected by the lysimeters used and represent an overestimate at the landscape scale. Instead writing that leaching exceeds precipitation one may also describe this as the ratio between leaching and precipitation.
b. In the reply to the first review the authors have written: "The 1305 total annual precipitation recorded could be an underestimate of this input at the plots especially because snowfall was not measured at the plot level but was used from a meteorological station 50km away from the site in Deer Lake, NL."
I do not see that this statement has been implemented in the manuscript and find the reporting of fluxes in its current form incorrect.

We have refined the annual precipitation estimate (1402mm) by using corrected Deer Lake Airport precipitation to fill a gap in our on-site precipitation data. See the new "water input estimate" section in the methods. Harvested water flux values still do exceed the annual water input, however discussion of this has been revised so as not to be misleading (see below and manuscript pg 12-13).

**c. Moreover, the reporting of precipitation is confusing.**
p. 3. L. 25 The region receives approximately 1095 mm of precipitation
on P. 9 L. 4 The regional mean annual air temperature over the July 2013 to July 2014 study period was + 3.4 °C (daily mean range: - 21.0 °C to + 22.7°C), and 1305 mm of total precipitation fell, including 483 mm water equivalents as snowfall.
I guess that the first values represents the long-term mean, while the second one represents the actual year – in the responses, I learnt that these measurements were taken 50 km away – this information has to be added.
The long-term regional means were removed from p.3 to limit confusion. A separate section was included ("water input estimate", page 5 lines 1-16) to clarify the use of the Deer Lake Airport Station (50km away) to correct our on site tipping bucket measurements. Annual estimates were revised and added to Table 1.

d. The collection of lateral flow has implications for the estimation of DOC fluxes. As the water fluxes in the O horizon exceed precipitation inputs, the area where DOC is coming from must exceed a m2. As other scientist may take the 38 and 54 g C/m2 (and the authors compare is with other studies and with net ecosystem C balances), I find that a critical discussion of this value is required.
The discussion has been revised to avoid direct comparison of our DOC flux values to other vertical fluxes of C (pg 11 line 13-30) and to explicitly state that lysimeters capture both vertical and lateral flow and the implications for landscape level modelling (pg 12 line 1 -29)
p.12 L. 20 ff the concept of DOC transport in boreal landscapes should be more strongly discussed by considering similar concepts that have been developed in other boreal regions, e.g. Sweden.
Water and DOC transport papers have been included in a discussion point regarding lateral transport (pg 12 line 20-25)
p. 12 L.25 The statement 'These models may be particularly inappropriate for use in highly porous surface organic soils and hillslopes with high precipitation rates as demonstrated here.' is principally correct. However, there are several hydrological models that include a macropore component (with their own uncertainty) and I find it impertinent to write such a statement and present the own data lysimeter based data rather uncritical and writing bluntly that water fluxes in O horizons exceed precipitation.
We have included a discussion of this on pg 12 lines 1-30. While macropore flow dynamics are described in water and solute transport models, organic horizons are not explicitly defined or included in these efforts despite having unique hydrologic properties and representing important sources of DOC. Measurement and incorporation of these properties, and the interaction with the underlying mineral soil, will help refine the landscape level role of organic horizons in DOC transport.

2. **Litterfall inputs** (mass and C) is inconsistent/wrong. The C input by litter hardly differs from the mass input (Table 1), but it should be approx.. 50% of it as C-concentration in litter are approx. 50%. The C input by litter is also reported at page 8, L. 11. I suppose that the mass provided in Table 1 is wrong but this has to be clarified.

Litterfall estimate was corrected and revised in manuscript

3. **Digits are reported incorrectly**. The reporting of digits is related to the accuracy of the measurement and estimation. In this manuscript, this is not applied, for instance:

Table 1 only 1 digit is given for soil moisture, which seems to coarse

Table 2: 3 decimals for statistical significance are sufficient

Figure 4: 4 decimals in the equations are too much

P. 8, L. 23. $CO_2$ flux, reported with 4 digits

Digits were corrected where indicated

4. **Minor comments**

- Statistical analyses: provide fixed and random effects in your lme-analysis to clarify that experimental design was captured.

Done

- p. 9 , L. 30 a 'and' is missing in the sentence

Done

- p.10, L. 19 I suggest to write 'measured water fluxes'

Done

- p.10, L. 21 replace 'corresponded'

Done

- Superscripts seem all written as subscripts

Done

[revised manuscript text omitted]

A. Site Layout

[Figure]

[Figure]

Downslope

North

B. Lysimeter Design

C. Lysimeter Field Installation and Measurement

**Figure 1. Pynn's Brook Experimental Forest Experimental Design.** A north facing black spruce hillslop site divided into 6 50x50m plots, half of which were randomly selected for harvest 10 years prior to lysimeter installation (A). Each plot contains two lysimeter pairs ("X") for a total of 12 harvest and 12 forest lysimeters. The lysimeters consisted of a HDPE tray with a sloped bottom connected to a funnel and PEX tubing (B). Each lysimeter was installed between the moss + organic and the mineral horizons on a slope ranging between 5 - 13%. Water collected by the lysimeters infilitrated vertically and laterally through moss and organic layers and into a 25 L reservoir from which samples were retrieved (C).

**Figure 2.** Temporal variation of soil respiration (a), daily mean soil temperature with the presence of a snowpack indicated by the grey (harvested) and black (forest) bar (b), daily rainfall and daily mean soil moisture (c), and lysimeter collections (d,e,f) from July 2013 to July 2014 in black spruce forest and harvested plots. The mean dissolved organic carbon (DOC) concentration (d), water flux (e), and DOC flux (f) was determined using passive pan lysimeter collections underneath O horizons. Lysimeter sampling was continuous and points represent a mean daily flux over each collection period. Error bars show standard error of the mean of 12 lysimeter collections per plot type per collection period. Grey shading areas indicate dry periods signified by those exceeding 10 consecutive days of rainfall less than 10mm/day, corresponding to periods of soil drying. Significant differences in DOC flux, water flux and DOC concentration between plot type on each collection day where determined by repeated measure linear mixed model post hoc tests and are indicated by an asterisk (alpha = 0.05).

[Figure]

[Figure]

p = 0.0903                    p = 0.0084                    p = 0.0337

**Figure 3**. Mean annual lysimeter collected variables. Volume weighted dissolved organic carbon (DOC) concentration (A), total water flux (B), and total DOC flux collected from organic horizons of forest (F) and harvested (H) plots over the entire study period. Annual values were calculated from the accumulated 29 sample collection time points taken from 12 F and 12 H passive pan lysimeters over one year from July 2013 to July 2014. Asterisks show significant differences between plot type (alpha = 0.05) determined using one-way plot nested ANOVA tests (Table S2).

[Figure]

**Figure 4.** Seasonal relationship between dissolved organic carbon (DOC) fluxes and water input to the soil in mature forest (F) and harvested (H) plots. Seasons are designated as summer (a), autumn (b) and winter + snowmelt (c).

[Figure]

**Figure 5.** Lysimeter Captured Water Fluxes versus Water Input over the Lysimeter Footprint in harvested (a) and forest (b) plots. Lysimeter collections made during periods when volumetric soil water content remained below soil matrix saturation (grey circles) contrast with lysimeter collections made during periods when soil matrix saturation was reached (black circles). Matrix saturation in harvested and forest plots was determined by infiltration experiments and complimented by soil evaportation measurements (see Table 4).